# Optimization of a Non-Traditional Vibration Absorber for Vibration Suppression and Energy Harvesting

**Miao Yuan [1], Youzuo Jin [1], Kefu Liu [1,\*] and Ayan Sadhu [2]**

1 Department of Mechanical Engineering, Lakehead University, Thunder Bay, ON P7B 5E1, Canada; myuan1@lakeheadu.ca (M.Y.); yjin12@lakeheadu.ca (Y.J.)
2 Department of Civil and Environmental Engineering, Western University, London, ON N6A 3K7, Canada; asadhu@uwo.ca
\* Correspondence: kliu@lakeheadu.ca

**Abstract:** This paper investigates the optimization of a non-traditional vibration absorber for simultaneous vibration suppression and energy harvesting. Unlike a traditional vibration absorber, the non-traditional vibration absorber has its damper connected between the absorber mass and the base. An electromagnetic energy harvester is used as a tunable absorber damper. This non-traditional vibration absorber is attached to a primary system that is subjected to random base excitation. An analytical study is conducted by assuming that the base excitation is white noise. In terms of vibration suppression, the objective of the optimization is to minimize the power dissipated by the primary damper and maximize the power dissipated by the absorber damper. It is found that when the primary system is undamped, the power dissipated by the absorber damper remains a constant that is related to the mass ratio. The higher the mass ratio, the higher the power dissipated. When the primary system is damped, the minimization of the power dissipated by the primary damping is equivalent to the maximization of the power dissipated by the absorber damper. The existence of the optimum solutions depends on both the mass ratio and the primary damping ratio. In terms of energy harvesting, the objective of optimization is to maximize the power harvested by the load resistor. It is found that for a given mass ratio and primary damping ratio, the optimum frequency tuning ratio required to maximize vibration suppression is slightly higher than that required to maximize the harvested power. The trade-off issue between vibration suppression and energy harvesting is investigated. An apparatus is developed to allow frequency tuning and damping tuning. Both the numerical simulation and experimental study with band-limited white noise validate the general trends revealed in the analytical study.

**Keywords:** non-traditional vibration absorber; vibration suppression; electromagnetic damper; energy harvesting

## 1. Introduction

Machines or structures are often subjected to vibrations, e.g., unbalanced rotating machines, vehicles on rough terrains, and high-rise buildings under wind or earthquake excitation. Such vibrations are undesirable because they generate noises, cause discomforts, result in fatigue failures and structural damages, etc. An undamped vibration absorber consists of a mass and spring. When a single degree-of-freedom primary system is subjected to a harmonic excitation, its steady state response can be completely suppressed by attaching the undamped vibration absorber that meets the tuning condition. However, the undamped vibration absorber has a narrow operating band, and its performance deteriorates significantly when the tuning condition is not satisfied. A damper can be added to form a damped vibration absorber or tuned mass damper (TMD) in order to improve the performance robustness. Traditionally a damper is added between the primary mass and absorber mass, as shown in Figure 1a where $m$, $k$, and $c$ are the mass, stiffness, and damping coefficient

of the primary system, respectively, $m_a$, $k_a$, and $c_a$ are the mass, stiffness, and damping coefficient of the absorber system, respectively, $y$ is the displacement of the base, $x$ and $x_a$ are the displacement of the primary mass and absorber mass relative to the base. Figure 1b shows a non-traditional way in which a damper is connected between the absorber mass and the base. The former is often referred to as "model A", while the latter "model B". Although model B is not as common as model A, it offers some advantages. For example, when the damper requires a certain stroke space in a tight space, the implementation of model B is easier than that of model A. For a pendulum-type TMD, model B may be the only viable option. When a TMD is used to control the resonance of a vibration isolator, placing the damper between the absorber mass and base reduces the amount of added mass to the isolated system.

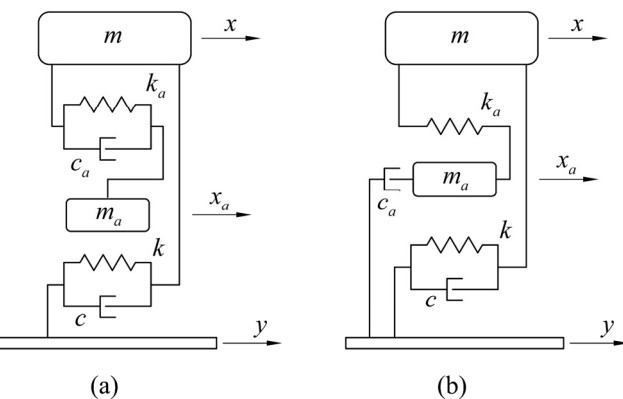

(a) (b)

**Figure 1.** Two configurations of TMD: (**a**) model A; (**b**) model B.

The two key parameters to design a TMD are the frequency tuning ratio $\beta$ and absorber damping ratio $\zeta_a$. The optimum design of a TMD intends to determine the best values of these two parameters in order to minimize the chosen objective function. Since the famous "fixed-points" theory was developed by Den Hartog [1], the optimum design of model A has been studied extensively. The study reported by Zilletti et al. [2] summarized the results. In what follows, the optimum design of model B is briefly reviewed. Ren [3] investigated the optimum design of model B attached to an undamped primary system subjected to a direct harmonic force excitation. Using the classical "fixed-points" theory, the optimum parameters were derived to minimize the maximum amplitude of the normalized displacement frequency response function (FRF). These results were later verified by Liu and Liu [4] using a more straightforward approach. Wong and Cheung [5] focused on the case of model B attached to an undamped primary system under a harmonic ground excitation. The objective function was chosen as the ratio between the absolute amplitude of the primary mass and that of the base harmonic motion. The same authors [6] further considered model B attached to an undamped primary system subjected to a direct harmonic force excitation. The objective function was defined as the ratio of the amplitude of the primary mass's absolute velocity and the amplitude of the base's velocity. Later, they [7] investigated the optimum design of model B attached to an undamped primary system subjected to random force excitation. The objective function was chosen to be the mean squared normalized FRF of the primary mass. In a follow-up study [8], the same authors revisited the problem that was first investigated by Ren [3] and Liu and Liu [4]. They showed that for harmonic force excitation, the optimum model B based on the "fixed-points" theory does not lead to the minimum resonant amplitude of the primary mass. They proposed a new optimum procedure that resulted in the optimum parameters that yield a lower maximum amplitude response.

Liu and Coppola [9] dealt with the case of model B attached to a damped primary system subjected to a direct force excitation. The objective function was the normalized displacement FRF. The approximate frequency tuning ratio was derived based on the "fixed-points" theory. The optimum damping ratio was found numerically. The same problem was

investigated by Anh and Nguyen [10]. The damped primary system was approximately replaced by an equivalent undamped one to apply the "fixed-points" theory. The analytical expressions for the optimum parameters were found. Xiang and Nishitani [11] addressed the issue of model B attached to a damped primary system subjected to a harmonic ground motion. The objective function was defined as the normalized absolute amplitude of the primary mass's acceleration. The design procedure was proposed such that the vibration suppression covers a wider frequency range than a traditional vibration absorber. The same authors [12] tackled the optimum design of the same model subjected to ground motion using the displacement coordinates relative to the ground. For harmonic ground excitation, the performance index was set as the maximum FRF magnitude of the relative displacement of the primary structure with respect to the ground acceleration and the derivation was based on the "fixed-points" theory. For transient responses, the stability maximization criterion (SMC) was used as the objective function for optimization. They also presented a summary of the previous study [13]. Notably, an experimental verification was conducted with a pendulum-type model B. The effect of frequency detuning on the performance of model B was investigated by Araz [14]. The numerical study showed that model B with a high mass ratio provides better robustness to the change in the target frequency ratio than model A.

Significant research has been conducted to harvest energy from ambient vibrations. Common practices of vibration energy harvesting include the use of piezoelectric materials [15] or electromagnetic devices [16]. As a typical energy harvester has a structure similar to a TMD, it is desirable to use it for a dual goal: vibration suppression (VS) and energy harvesting (EH). The following literature review will focus on simultaneous VS and EH using TMD. Ouled Chtiba et al. [17] optimized the locations, masses, stiffnesses, and damping coefficients of a set of TMDs to minimize the total energy of the primary structure. Later, they [18] proposed an optimal design for a set of TMDs incorporated with piezoelectric devices. Their analytical results were verified by computer simulation. A survey of control strategies for VS and EH via piezoceramics was presented by Wang and Inman [19]. A TMD proposed by Tang and Zuo [20] consisted of a rotary motor with gears and a rack-pinion mechanism. By connecting the output of the motor to a resistive load, the system functions as an electromagnetic damper. The study reported by Harne [21] compared the performance of an electromagnetic energy harvester and that of a piezoelectric energy harvester when each of them was coupled to a mass-spring-damper system. The classical vibration absorber analysis was extended by including electromechanical damping and stiffening effects introduced by the energy harvester. The EH TMD proposed by Ali and Adhikari [22] consisted of a mass–spring–damper system coupled with a piezoelectric element. The "fixed-points" theory was approximately applied to find the optimum parameters. Simultaneous VS and EH were investigated by Brennan et al. [23]. Two types of vibration were considered: broadband excitation and single frequency excitation. The performances of VS and EH were analyzed by using several criteria. The electromagnetic vibration absorber proposed by Gonzalez-Buelga et al. [24] possessed both energy recovery and frequency tuning capabilities. An energy regenerative TMD reported by Shen et al. [25] consisted of a pendulum, an electromagnetic damper, and an EH circuit. A hybrid energy harvester was proposed [22,26] in which a piezoelectric harvester was attached between the primary mass and the ground while an electromagnetic harvester was between the absorber mass and ground. However, the model for the electromagnetic absorber still belonged to model A as the damping force of the electromagnetic harvester was proportional to the displacement of the absorber relative to the primary mass. An experimental study was conducted by Zoka and Afsharfard [27] with a so-called double stiffness vibration suppressor and energy harvester. Multi-objective optimization was carried out by defining a combined objective function with the concept of the perfection rate. A novel geometric approach was proposed [28] for the optimal design of simultaneous VS and EH of TMDs. Kecik and Mitura [29] developed a pendulum TMD equipped with a magnetic rotatory harvester and a maglev harvester. The effectiveness of EH and VS of both harvesters was

compared. The TMD proposed by Ali and Adhikari [22] was used to study parametric uncertainty and random excitation [30]. For the random excitation, the mean squared value of normalized primary mass relative displacement was used as the objective function for optimum VS and the mean squared value of normalized generated voltage was used as the objective function for optimum EH.

It should be noted that the aforementioned studies are based on model A. Recently an electromagnetic model B was developed for the dual purpose [31]. The device was attached to a primary system subjected to a harmonic base excitation. The relative displacement transmissibility ratio was chosen to be the objective function for optimum VS, while the ratio of the harvested power amplitude to the input power amplitude was chosen to be the objective function for optimum EH. The same device was used in a primary system under initial disturbance [32]. The SMC was employed as the objective function for optimum VS, while the ratio of the harvested energy over the input energy was used as the objective function for optimum EH. The difference between the present study and these two reported ones is two-fold. First, the type of excitation is different as the present study deals with random base excitation. Second, the objective functions for optimization are different. The objective for VS is to minimize the power dissipated by the primary damping and maximize the power dissipated by the absorber damping. The objective for EH is to maximize the harvested power by the load resistor. To the best of our knowledge, the novelty of the present study lies in the three aspects. First, model B, other than model A, is used to achieve simultaneous VS and EH under random ground excitation. Second, the optimum criteria are based on both VS and EH. Third, as the model under consideration is defined using the relative coordinates and the mean squared relative velocity of the primary mass is used as the objective function, the optimum results are different from those reported by Cheung and Wong [8].

The remainder of the paper is organized as follows. Section 2 addresses the optimization of a model B TMD for VS. Section 3 discusses optimization of a model B TMD for EH. Section 4 presents result validation using numerical simulation and experiment. Section 5 draws the conclusion of this study.

## 2. Optimization of a Non-Traditional TMD for Vibration Suppression

The equations of motion for model B shown in Figure 1b are given by:

$$\ddot{x} + \left(\omega_p^2 + \mu\omega_a^2\right)x - \mu\omega_a^2 x_a + 2\zeta_p\omega_p\dot{x} = -\ddot{y}$$
$$\ddot{x}_a + 2\zeta_a\omega_p\dot{x}_a + \omega_a^2 x_a - \omega_a^2 x = -\ddot{y} \tag{1}$$

where

$$\omega_p = \sqrt{\frac{k}{m}}, \ \omega_a = \sqrt{\frac{k_a}{m_a}}, \ \mu = \frac{m_a}{m}, \ \zeta_p = \frac{c}{2m\omega_p}, \ \zeta_a = \frac{c_a}{2m_a\omega_p}$$

The notations $\omega_p$ and $\omega_a$ are the natural frequencies of the primary system and absorber system, respectively, $\mu$ is referred to as the mass ratio, $\zeta_p$ and $\zeta_a$ are the primary damping ratio and absorber damping ratio, respectively. To find the velocity FRFs of the primary mass and absorber mass, the ground motion is first assumed to be harmonic or $y = Ye^{j\omega t}$ where $j = \sqrt{-1}$, $Y$ and $\omega$ are the amplitude and frequency of the base motion, respectively. Then, the steady-state responses of the primary mass and TMD's mass are defined by $x = Xe^{j\omega t}$ and $x_a = X_a e^{j\omega t}$, respectively. Substituting $y$, $x$, and $x_a$ into Equation (1) results in

$$\omega_p^2 \begin{bmatrix} 1 - r^2 + \mu\beta^2 + j2\zeta_p r & -\mu\beta^2 \\ -\beta^2 & \beta^2 - r^2 + j2\zeta_a r \end{bmatrix} \begin{bmatrix} X \\ X_a \end{bmatrix} = \begin{bmatrix} 1 \\ 1 \end{bmatrix}\omega^2 Y \tag{2}$$

where $r = \omega/\omega_p$ and $\beta = \omega_a/\omega_p$ are referred to as the frequency ratio and frequency tuning ratio, respectively. Solving Equation (2) results in

$$X = \frac{(1+\mu)\beta^2 - r^2 + j2\zeta_a r}{\omega_p^2 \left[(1-r^2)(\beta^2 - r^2) - (\mu\beta^2 + 4\zeta_p\zeta_a)r^2\right] + j2r[(\beta^2 - r^2)\zeta_p + (1 - r^2 + \mu\beta^2)\zeta_a]} \omega^2 Y \tag{3}$$

and

$$X_a = \frac{(1+\mu)\beta^2 + 1 - r^2 + j2\zeta_p r}{\omega_p^2 \left[(1-r^2)(\beta^2 - r^2) - (\mu\beta^2 + 4\zeta_p\zeta_a)r^2\right] + j2r[(\beta^2 - r^2)\zeta_p + (1 - r^2 + \mu\beta^2)\zeta_a]} \omega^2 Y \tag{4}$$

As the steady-state relative velocities are given by $\dot{x} = j\omega X e^{j\omega t} = V e^{j\omega t}$ and $\dot{x}_a = j\omega X_a e^{j\omega t} = V_a e^{j\omega t}$, the amplitudes of the steady-state relative velocities are defined by

$$V = \frac{-2\zeta_a r^2 + j[(1+\mu)\beta^2 - r^2]r}{\omega_p \left[(1-r^2)(\beta^2 - r^2) - (\mu\beta^2 + 4\zeta_p\zeta_a)r^2\right] + j2r[(\beta^2 - r^2)\zeta_p + (1 - r^2 + \mu\beta^2)\zeta_a]} \omega^2 Y \tag{5}$$

and

$$V_a = \frac{-2\zeta_p r^2 + j[(1+\mu)\beta^2 + 1 - r^2]r}{\omega_p \left[(1-r^2)(\beta^2 - r^2) - (\mu\beta^2 + 4\zeta_p\zeta_a)r^2\right] + j2r[(\beta^2 - r^2)\zeta_p + (1 - r^2 + \mu\beta^2)\zeta_a]} \omega^2 Y \tag{6}$$

The velocity FRF of the primary mass is defined by

$$G(r) = \frac{V}{\omega^2 Y / \omega_p} = \frac{-2\zeta_a r^2 + jr[(1+\mu)\beta^2 - r^2]}{\left[(1-r^2)(\beta^2 - r^2) - (\mu\beta^2 + 4\zeta_p\zeta_a)r^2\right] + 2jr[(\beta^2 - r^2)\zeta_p + (1 + \mu\beta^2 - r^2)\zeta_a]} \tag{7}$$

and the velocity FRF of the absorber mass is defined by

$$G_a(r) = \frac{V_a}{\omega^2 Y / \omega_p} = \frac{-2\zeta_p r^2 + jr[(1+\mu)\beta^2 + 1 - r^2]}{\left[(1-r^2)(\beta^2 - r^2) - (\mu\beta^2 + 4\zeta_p\zeta_a)r^2\right] + 2j\mu r[(\beta^2 - r^2)\zeta_p + (1 + \mu\beta^2 - r^2)\zeta_a]} \tag{8}$$

The power spectral density (PSD) function of the base excitation is defined as

$$S(\omega) = \frac{1}{2\pi}|A(\omega)|^2 \tag{9}$$

where $A(\omega)$ is the Fourier transform of the base acceleration function [33]. Now, assume that the ground motion is white noise with a uniform PSD function of $S_0$. To measure the response magnitudes, the mean squared relative velocity [2] is used. The mean squared relative velocity $E(V^2)$ of the primary mass is defined by

$$E(V^2) = \int_{-\infty}^{\infty} V^2 d\omega = \frac{S_0}{\omega_p} \int_{-\infty}^{\infty} |G(r)|^2 dr \tag{10}$$

and the mean squared relative velocity $E(V_a^2)$ of the absorber mass is defined by

$$E(V_a^2) = \int_{-\infty}^{\infty} V_a^2 d\omega = \frac{S_0}{\omega_p} \int_{-\infty}^{\infty} |G_a(r)|^2 dr \tag{11}$$

For this study, two performance indices are defined as below

$$I_1 = \frac{1}{2}c\frac{E[V^2]}{2\pi S_0 m} \tag{12}$$

and

$$I_2 = \frac{1}{2}c_a \frac{E\left[V_a^2\right]}{2\pi S_0 m} \tag{13}$$

In the above equations, the constant $2\pi S_0 m$ is introduced to ensure the indices are dimensionless. Apparently, the first index $I_1$ measures the power dissipated by the primary damping, while the second index $I_2$ the power dissipated by the absorber damping. Substituting Equation (10) into Equation (12) results in

$$I_1 = \frac{\zeta_p}{2\pi} \int_{-\infty}^{\infty} |G(r)|^2 dr \tag{14}$$

Substituting Equation (11) into Equation (13) yields

$$I_2 = \frac{\mu \zeta_a}{2\pi} \int_{-\infty}^{\infty} |G_a(r)|^2 dr \tag{15}$$

Following the procedures given by Newland [34], the integrals in Equations (14) and (15) can be evaluated. Thus, the defined indices can be expressed explicitly as

$$I_1 = \frac{\zeta_p}{4} \frac{A_1\beta^4 + A_2\beta^2 + A_3}{B_1\beta^4 + B_2\beta^2 + B_3} \tag{16}$$

$$I_2 = \frac{\zeta_a \mu}{4} \frac{A_1\beta^4 + A_4\beta^2 + A_5}{B_1\beta^4 + B_2\beta^2 + B_3} \tag{17}$$

The constants in Equations (16) and (17) are given in Appendix A.

First, an undamped primary system is considered or $\zeta_p = 0$. In this case, the indices defined in Equations (16) and (17) become

$$I_1 = 0 \tag{18}$$

$$I_2 = \frac{\mu + 1}{4} \tag{19}$$

Equation (18) is self-explanatory as no power is dissipated by a damping-free primary system. Equation (19) indicates that the power dissipated by the absorber damper is constant. This can be explained by the energy balance. When no damping exists in the primary system, the input power equals to the power dissipated by the absorber damping. The input power is constant as the excitation is assumed to be ideally white. As a matter of fact, if both $\zeta_p$ and $\zeta_a$ are not zero, the sum of $I_1$ and $I_2$ is also equal to the constant given by Equation (19), as proven in Appendix A.

To investigate the optimum frequency tuning ratio $\beta^*$ and damping ratio $\zeta_a^*$ that minimizes the mean squared relative velocity of the primary mass, the following index is defined

$$I_p = \frac{1}{2\pi} \int_{-\infty}^{\infty} |G|^2 dr = \frac{1}{4} \frac{(1+\mu)\beta^4 + 2\left(2\mu\zeta_a^2 - 1\right)\beta^2 + 4\zeta_a^2 + 1}{\mu\zeta_a\beta^4} \tag{20}$$

The following two conditions should be satisfied

$$\frac{\partial I_p}{\partial \beta} = 0 \tag{21}$$

$$\frac{\partial I_p}{\partial \zeta_a} = 0 \tag{22}$$

which result in

$$\left(2\mu\beta^2 + 4\right)\zeta_a^2 - \beta^2 + 1 = 0 \tag{23}$$

$$4\left(\mu\beta^2+1\right)\zeta_a^2 - (1+\mu)\beta^4 + 2\beta^2 - 1 = 0 \tag{24}$$

Equation (23) yields

$$\zeta_a = \sqrt{\frac{\beta^2-1}{2\mu\beta^2+4}}, \ \beta > 1 \tag{25}$$

Substituting Equation (25) into Equation (24) results in

$$(1+\mu)\mu\beta^6 + 2(1-\mu)\beta^4 + 3(\mu-2)\beta^2 + 4 = 0 \tag{26}$$

For Equation (26) to have real positive roots, the following condition needs to be satisfied

$$\Delta = 4 - 27\mu^5 - 72\mu^4 - 184\mu^3 - 87\mu^2 - 12\mu > 0 \tag{27}$$

which holds true only when $\mu < 0.142$. Table 1 lists the optimum parameters obtained by solving Equation (26) first and Equation (25) next. The results show that with an increase in the mass ratio, the optimum frequency tuning ratio and damping ratio need to increase as well.

**Table 1.** The optimum parameters that minimize $I_p$ defined by Equation (20).

| $\mu$ | $\beta^*$ | $\zeta_a^*$ | $I_p$ |
|-------|-----------|-------------|-------|
| 0.05 | 1.028 | 0.119 | 4.466 |
| 0.06 | 1.035 | 0.132 | 4.074 |
| 0.07 | 1.043 | 0.145 | 3.768 |
| 0.08 | 1.051 | 0.158 | 3.521 |
| 0.09 | 1.059 | 0.171 | 3.316 |
| 0.10 | 1.069 | 0.184 | 3.141 |
| 0.11 | 1.081 | 0.199 | 2.989 |
| 0.12 | 1.095 | 0.215 | 2.855 |
| 0.13 | 1.113 | 0.235 | 2.736 |
| 0.14 | 1.146 | 0.268 | 2.626 |

Figure 2 shows the contours (solid line) of $I_p$ and the solution (dashed line) of Equation (25) when $\mu = 0.05$. The local minimum point is marked by a box, while the local maximum point is marked by a circle. As shown in the figure, the variation in the mean squared relative velocity is relatively small at any point on the dashed line. A similar phenomenon was observed when the mean squared displacement was minimized for model B subjected to a direct force excitation [7]. Figure 3 shows the results of $\mu = 0.15$. In this case, no local minimum point existed as Equation (27) is no longer satisfied.

Now the damped primary system is considered. For the sake of VS, the power dissipated by the primary damping should be minimized, while the power dissipated by the absorber damping should be maximized. Thus, the following conditions must be satisfied,

$$\frac{\partial I_1}{\partial\beta} = 0, \ \frac{\partial I_1}{\partial\zeta_a} = 0 \tag{28}$$

$$\frac{\partial I_2}{\partial\beta} = 0, \ \frac{\partial I_2}{\partial\zeta_a} = 0 \tag{29}$$

Substituting Equation (16) into Equation (28) yields,

$$\left(\mu^2\zeta_p^2 + 2\mu\zeta_p\zeta_a - 2\mu^2\zeta_a^3 + 2\mu^2\zeta_p^3\zeta_a + \zeta_a^2 - 2\mu\zeta_a^4 + 2\mu\zeta_p^4 - 2\zeta_p\zeta_a^3 + 2\zeta_p^3\zeta_a\right)\beta^4 + \left(\mu\zeta_p^2 + 4\zeta_p^2\zeta_a^2 - 4\mu\zeta_p\zeta_a^3 + \zeta_p\zeta_a - \mu\zeta_p\zeta_a - \zeta_a^2 - 4\zeta_a^4 + 4\mu\zeta_p^3\zeta_a\right)\beta^2 - \zeta_p\zeta_a - 4\zeta_p^2\zeta_a^2 - 4\zeta_p\zeta_a^3 = 0 \tag{30}$$

$$\left(-4\beta^2-4\mu\beta^4\right)\zeta_a^4 + \left(8\zeta_p\mu\beta^4 - 16\zeta_p - 24\zeta_p\mu\beta^2 + 8\zeta_p\beta^2 - 8\zeta_p\mu^2\beta^4\right)\zeta_a^3 + \left(\beta^6 + 8\zeta_p^2\mu^2\beta^4 + \mu\beta^6 - 8\zeta_p^2 - 2\beta^4 + 4\zeta_p^2\beta^4 + \beta^2\right)\zeta_a^2 + \left(2\zeta_p\mu\beta^6 + 8\zeta_p^3\mu\beta^2 + 8\zeta_p^3\mu^2\beta^4 + 2\zeta_p\mu^2\beta^6 + 8\zeta_p^3\mu\beta^4\right)\zeta_a + \zeta_p^2\mu^2\beta^6 + 4\zeta_p^4\mu\beta^4 + \zeta_p^2\mu\beta^2 + \zeta_p^2\mu^3\beta^6 + 2\zeta_p^2\mu^2\beta^4 = 0 \tag{31}$$

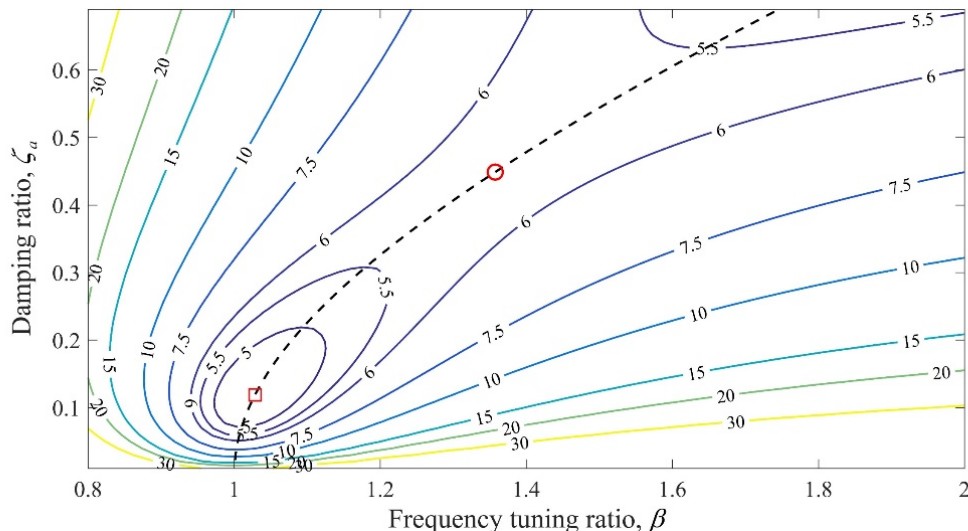

**Figure 2.** Contour plot of $I_p$ superimposed with the solution (black dashed line) of Equation (25) when $\mu = 0.05$.

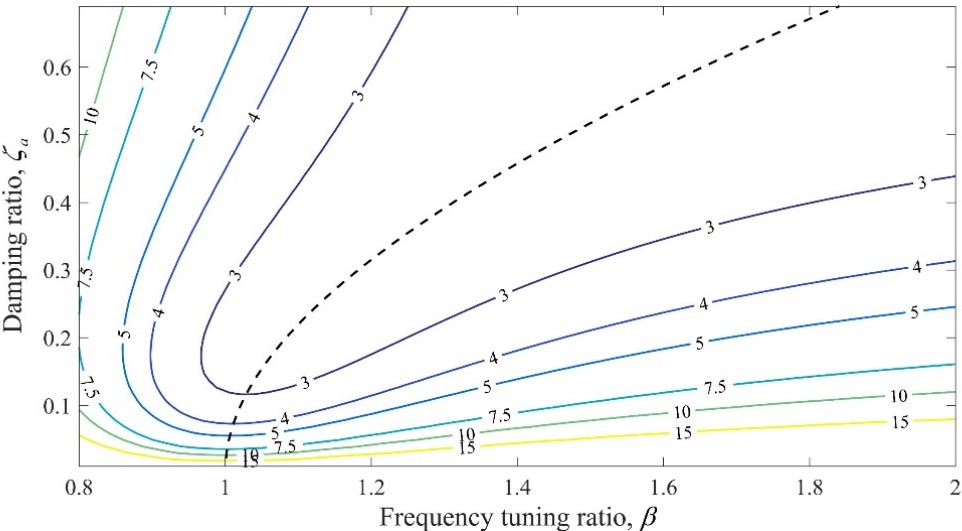

**Figure 3.** Contour plot of $I_p$ superimposed with the solution (black dashed line) of Equation (25) when $\mu = 0.15$.

Substituting Equation (17) into Equation (29) yields the same set of equations. This indicates that minimizing the power dissipated by the primary damping is equivalent to maximizing the power dissipated by the absorber damping for the same set of $\mu$ and $\zeta_p$. The simultaneous solution of the above equations for $\beta$ and $\zeta_a$ can be mathematically challenging. In this study, a different approach was employed. When a set of the values of $\mu$ and $\zeta_p$ is given, a real positive $\beta$ value can be numerically obtained by solving Equation (30) by specifying an $\zeta_a$ value. By varying the $\zeta_a$ value incrementally, this procedure was repeated for the same set of the values of $\mu$ and $\zeta_p$. Eventually, the relationship of $\zeta_a$ verses $\beta$ could be established. Using the same set of $\mu$ and $\zeta_p$, the relationship of $\beta$ verses $\zeta_a$ could be established by solving Equation (31) by specifying a $\beta$ value. Figure 4 shows the contour plot of $I_1$ and the solutions of Equations (30) and (31) when $\mu = 0.05$ and $\zeta_p = 0.05$. The two

solutions intersected at two locations. The intersection marked by a box is a local minimum point, while the intersection marked by a circle is a local maximum point.

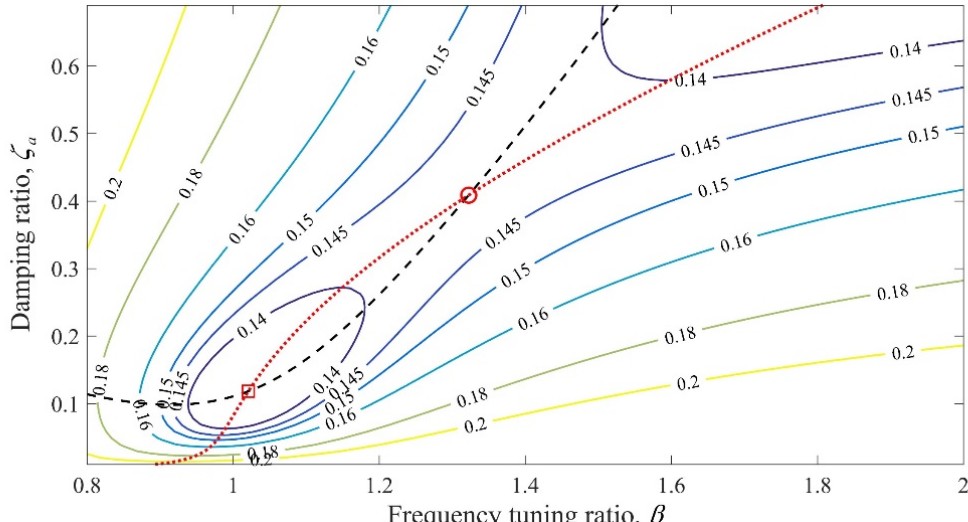

**Figure 4.** Contour plot of $I_1$ superimposed with the solution (black dashed line) of Equation (30) and the solution (red dotted line) of Equation (31) when $\mu = 0.05$ and $\zeta_p = 0.05$.

Figure 5 shows the results for the case of $\mu = 0.15$ and $\zeta_p = 0.05$. Now, the two curves did not intersect one another, indicating that a local minimum did not exist. Table 2 lists the optimum parameters for the two different primary damping ratios. For a fixed primary damping ratio, the optimum frequency tuning ratio and absorber damping ratio increased with an increase in the mass ratio. This indicates that higher damping is required to dissipate the energy if a larger mass ratio is used. It can also be seen that for a fixed mass ratio, a higher frequency tuning ratio and damping ratio were required for a system with a smaller primary damping ratio. This is expected as there is more energy required to be dissipated by the absorber damper. No optimum solution existed when $\mu > 0.127$ for the case of $\zeta_p = 0.05$ while no optimum solution existed when $\mu > 0.131$ for the case of $\zeta_p = 0.005$.

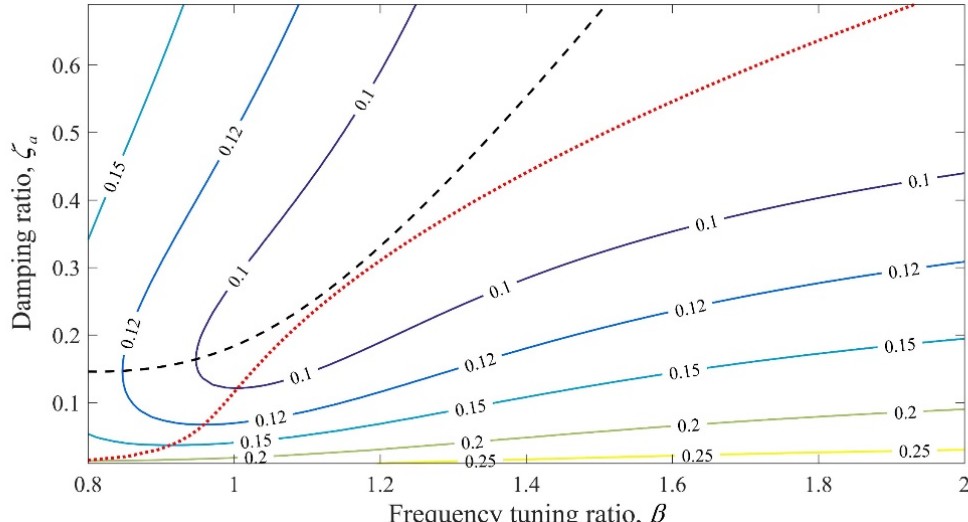

**Figure 5.** Contour plot of $I_1$ superimposed with the solution (black dashed line) of Equation (30) and the solution (red dotted line) of Equation (31) when $\mu = 0.15$ and $\zeta_p = 0.05$.

**Table 2.** The optimum parameters for VS and the corresponding optimum $I_1$ values when $\zeta_p = 0.05$ or $\zeta_p = 0.005$.

| | $\zeta_p$=0.05 | | | $\zeta_p$=0.005 | | |
|---|---|---|---|---|---|---|
| $\mu$ | $\beta^*$ | $\zeta_a^*$ | $I_1^*$ | $\beta^*$ | $\zeta_a^*$ | $I_1^*$ |
| 0.05 | 1.021 | 0.118 | 0.131 | 1.028 | 0.119 | 0.021 |
| 0.06 | 1.027 | 0.132 | 0.125 | 1.035 | 0.132 | 0.019 |
| 0.07 | 1.034 | 0.145 | 0.119 | 1.042 | 0.145 | 0.018 |
| 0.08 | 1.042 | 0.158 | 0.114 | 1.050 | 0.158 | 0.017 |
| 0.09 | 1.051 | 0.171 | 0.110 | 1.059 | 0.171 | 0.016 |
| 0.10 | 1.062 | 0.185 | 0.106 | 1.069 | 0.185 | 0.015 |
| 0.11 | 1.075 | 0.201 | 0.103 | 1.081 | 0.199 | 0.014 |
| 0.12 | 1.092 | 0.220 | 0.100 | 1.095 | 0.216 | 0.014 |
| 0.13 | – | – | – | 1.114 | 0.236 | 0.013 |

The sensitivity of $I_1$ to the detuning of the frequency tuning ratio $\beta^*$ or the absorber damping ratio $\zeta_a^*$ was investigated. Two sets of the optimum parameter values corresponding to $\mu = 0.05$ and $\mu = 0.10$ were examined. By fixing the mass ratio as $\mu = 0.05$ or $\mu = 0.10$, the optimum frequency tuning ratio $\beta^*$ or $\zeta_a^*$ was varied by $\pm 10\%$, respectively. The variability is measured by

$$\Delta I_1 = \frac{I_1 - I_1^*}{I_1^*} \times 100\% \tag{32}$$

where $I_1$ is the value responding to the parameter set with one detuned parameter. Table 3 lists the results. Three observations can be drawn. First, all variations were positive no matter whether the optimum parameter was increased or decreased. This further validated that the optimum frequency tuning ratio and the optimum absorber damping ratio yield a minimum $I_1$ value. Second, the $I_1$ value was more sensitive to the variation in the frequency tuning ratio than to the variation in the absorber damping ratio. This is understandable as the frequency tuning ratio is the key parameter that affects the functionality of the vibration absorber. Third, the $I_1$ value was more sensitive to detuning if the primary damping was low. This can be explained by the fact that high primary damping facilitates the energy dissipation.

**Table 3.** Variation in $I_1$ when $\beta^*$ or $\zeta_a^*$ is varied by $\pm 10\%$.

| | $\zeta_p$=0.05 | | | $\zeta_p$=0.005 | | |
|---|---|---|---|---|---|---|
| $\mu$ | $\beta$ | $\zeta_a$ | $\Delta I_1$ (%) | $\beta$ | $\zeta_a$ | $\Delta I_1$ (%) |
| 0.05 | 1.123 | 0.118 | 7.9 | 1.131 | 0.119 | 23.02 |
| 0.05 | 0.919 | 0.118 | 11.0 | 0.925 | 0.119 | 40.08 |
| 0.05 | 1.021 | 0.130 | 0.19 | 1.028 | 0.131 | 0.41 |
| 0.05 | 1.021 | 0.107 | 0.23 | 1.028 | 0.107 | 0.50 |
| 0.10 | 1.115 | 0.185 | 4.91 | 1.123 | 0.185 | 10.26 |
| 0.10 | 1.009 | 0.185 | 7.50 | 1.016 | 0.185 | 18.29 |
| 0.10 | 1.062 | 0.195 | 0.24 | 1.069 | 0.194 | 0.42 |
| 0.10 | 1.062 | 0.176 | 0.29 | 1.069 | 0.175 | 0.52 |

### 3. Optimization of a Non-Traditional TMD for Energy Harvesting

In this section, optimization of a non-traditional TMD for EH is considered. Assuming that the absorber damper is an electromagnetic energy harvester similar to the one employed by Yuan et al. [31], its damping coefficient can be approximated by

$$c_a = \frac{\Theta^2}{2R_{coil} + R_{load}} \tag{33}$$

where $\Theta$ is the transduction factor, $R_{coil}$ is the resistance of one coil, and $R_{load}$ is the resistance of the load resistor. Equation (33) indicates that the absorber's damping coefficient is inversely proportional to the load resistance for a given electromagnetic energy harvester. The instantaneous power dissipated by the absorber damper can be computed by

$$p_e(t) = c_a \dot{x}_a^2(t) = \frac{\Theta^2}{2R_{coil} + R_{load}} \dot{x}_a^2(t) \tag{34}$$

On the other hand, the instantaneous power harvested by the load resistor is given by

$$p_{load}(t) = R_{load} i^2(t) \tag{35}$$

where $i$ is the current of the circuit. The inductance of the coils is neglected for simplicity [32]. With this condition, the current is related to the absorber mass's velocity relative to the base, i.e.,

$$i = \frac{\Theta}{2R_{coil} + R_{load}} \dot{x}_a \tag{36}$$

Substituting Equation (36) into Equation (35) yields

$$p_{load}(t) = \frac{R_{load}}{2R_{coil} + R_{load}} c_a \dot{x}_a^2(t) \tag{37}$$

A term can be defined as follows

$$f(R_{load}) = \frac{p_{load}(t)}{p_e(t)} = \frac{R_{load}}{2R_{coil} + R_{load}} \tag{38}$$

This term represents the percentage of power available for harvesting from the power dissipated by the electromagnetic energy harvester. Equation (38) shows that an increase in the load resistance results in an increase in the harvested power percentage. This conflicts with the goal of VS that requires high damping by lowering load resistance. Apparently, there is a trade-off between VS and EH.

Based on Equations (35) and (36), the accumulated energy harvested by the load resistor is given by

$$E_{load} = \frac{R_{load}}{2R_{coil} + R_{load}} c_a \int_0^\infty \dot{x}_a^2(t) dt \tag{39}$$

Comparing Equation (39) with Equation (11) indicates that the following index can be used to measure the EH efficiency,

$$I_3 = \frac{R_{load}}{2R_{coil} + R_{load}} \frac{\mu \zeta_a}{2\pi} \int_{-\infty}^\infty |G_a|^2 dr = \frac{R_{load}}{2R_{coil} + R_{load}} I_2 \tag{40}$$

Numerical investigations were conducted with the following system parameters: $m = 0.34$ kg, $f_p = 13.7$ Hz, $\Theta = 2.596$ Tm, and $R_{coil} = 2.3$ $\Omega$, where $f_p = \omega_p / (2\pi)$. Figure 6 shows the contours of $I_2$ and the contours of $I_3$ for the case of $\mu = 0.05$ and $\zeta_p = 0.05$. The solutions of Equations (30) and (31) were also given where their intersection marked by a box was a local maximum point for $I_2$ that corresponded to $\beta^* = 1.020$ and $R_{load}^* = 15.0$ $\Omega$. In Figure 6b, a local maximum of $I_3$ is marked by an asterisk and corresponded to $\beta^* = 0.9942$ and $R_{load}^* = 27.5$ $\Omega$. This confirms the trade-off issue mentioned above. Figure 7 shows the

results for the case of $\mu = 0.05$ and $\zeta_p = 0.005$. Now the optimum parameters to maximize $I_2$ became $\beta^* = 1.027$ and $R^*_{load} = 15.0\ \Omega$, while the optimum parameters to maximize $I_3$ became $\beta^* = 1.000$ and $R^*_{load} = 52.5\ \Omega$. With a decrease in the primary damping ratio, the energy dissipated by the absorber damping increases, and the load resistance should be reduced in order to reduce the energy consumed by the coil resistance.

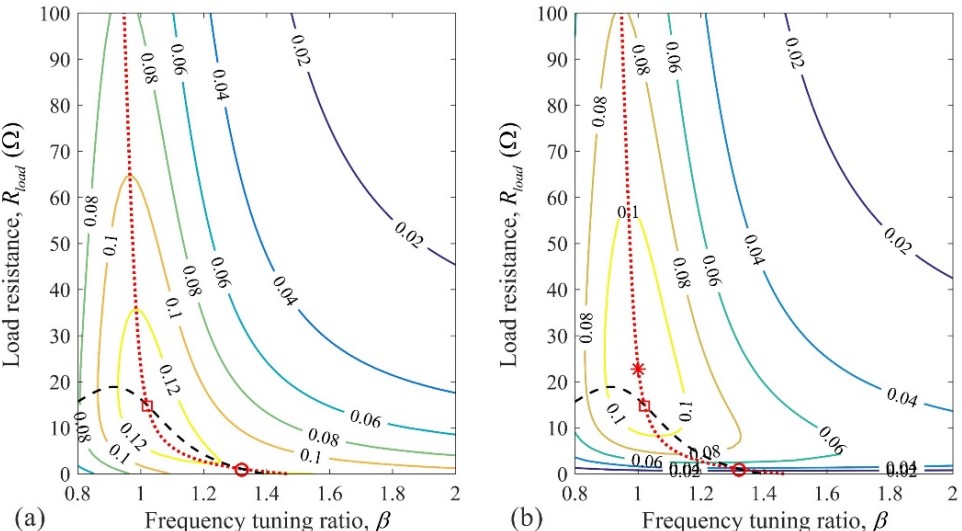

**Figure 6.** Contour plots of $I_2$ (**a**) and $I_3$ (**b**) superimposed with Equation (30)'s solution (black dashed line) and Equation (31)'s solution (red dotted line) when $\mu = 0.05$ and $\zeta_p = 0.05$.

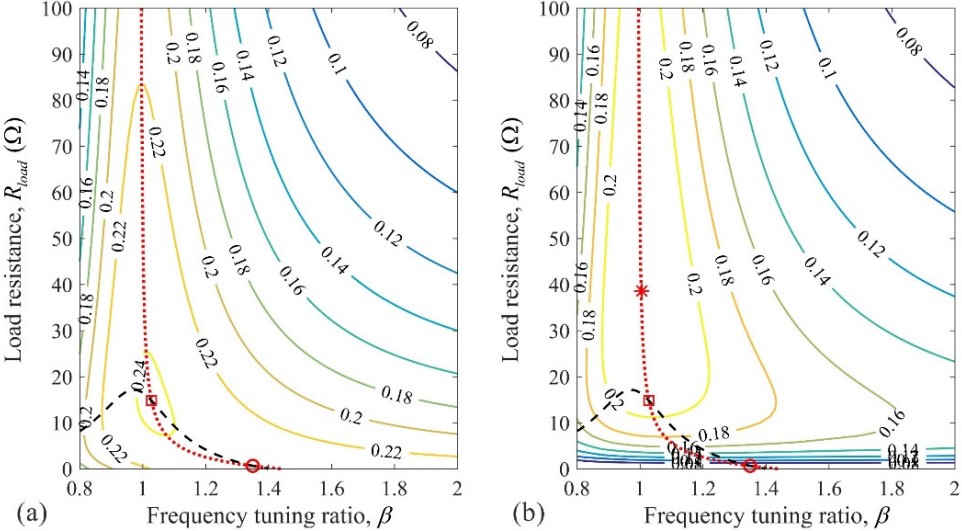

**Figure 7.** Contour plots of $I_2$ (**a**) and $I_3$ (**b**) superimposed with Equation (30)'s solution (black dashed line) and Equation (31)'s solution (red dotted line) when $\mu = 0.05$ and $\zeta_p = 0.005$.

Table 4 lists the optimum parameters that maximized the index $I_3$ for the two different primary damping ratios. It can be seen that the frequency tuning ratio varied little regardless of the mass ratio and primary damping ratio. For a fixed primary damping ratio, the optimum load resistance decreased with an increase in the mass ratio. For a fixed mass ratio, the optimum load resistance increased with a decrease in the primary damping ratio. By increasing $\mu$, one can continue to find the optimum parameters that maximize $I_3$. For example, when $\mu = 0.25$, the optimum solutions became $\beta^* = 0.83$ and $R^*_{load} = 12.5\ \Omega$. However, a local maximum no longer existed for $I_2$.

**Table 4.** The optimum parameters for EH when $\zeta_p = 0.05$.

| $\mu$ | $\zeta_p$=0.05 | | $\zeta_p$=0.005 | |
| | $\beta^*$ | $R^*_{load}(\Omega)/\zeta^*_a$ | $\beta^*$ | $R^*_{load}(\Omega)/\zeta^*_a$ |
|---|---|---|---|---|
| 0.05 | 0.994 | 27.5/0.072 | 1.000 | 52.5/0.040 |
| 0.06 | 0.989 | 23.3/0.069 | 0.999 | 46.7/0.037 |
| 0.07 | 0.984 | 20.0/0.067 | 0.998 | 42.5/0.035 |
| 0.08 | 0.978 | 18.3/0.063 | 0.996 | 40.0/0.032 |
| 0.09 | 0.972 | 16.7/0.060 | 0.995 | 37.5/0.030 |
| 0.10 | 0.965 | 15.8/0.056 | 0.993 | 35.8/0.028 |
| 0.11 | 0.958 | 15.0/0.053 | 0.992 | 34.2/0.027 |
| 0.12 | 0.951 | 14.2/0.051 | 0.990 | 33.3/0.025 |
| 0.13 | 0.945 | 13.3/0.049 | 0.988 | 31.7/0.024 |

Figure 8 shows the results for the case of $\mu = 0.08$ and $\zeta_p = 0.05$. As indicated by a box in Figure 8a, the optimum solution in terms of VS became $\beta^* = 1.05$ and $R^*_{load} = 4.167\ \Omega$. This indicates that with a higher mass ratio, a higher absorber damping ratio or a lower load resistance is required to achieve a greater amount of the dissipated power. However, indicated by an asterisk in Figure 8b, the optimum solution in terms of EH became $\beta^* = 0.98$ and the load resistance of $R^*_{load} = 18.333\ \Omega$. Now the trade-off between VS and EH becomes more apparent. Reducing the load resistance enhances VS and compromises EH as more energy is dissipated by the coil resistance.

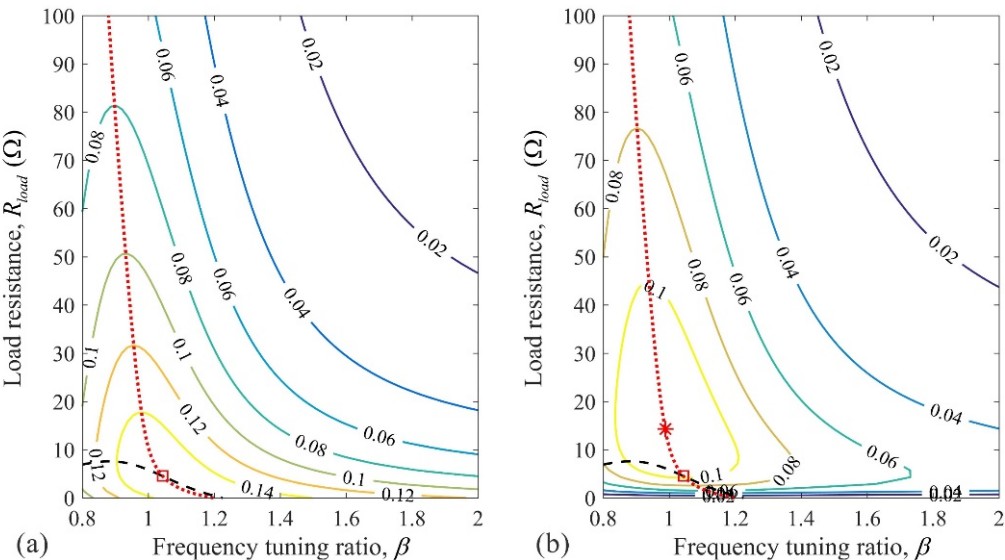

**Figure 8.** Contour plots of $I_2$ (**a**) and $I_3$ (**b**) superimposed with Equation (30)'s solution (black dashed line) and Equation (31)'s solution (red dotted line) when $\mu = 0.08$ and $\zeta_p = 0.05$.

On the other hand, an increase in the primary damping had little effect on the trade-off situation. Figure 9 shows for the case of $\mu = 0.08$ and $\zeta_p = 0.15$. The optimum solution in terms of VS became $\beta^* = 1.003$, $R^*_{load} = 5.0\ \Omega$, $I_2 = 0.0771$, and $I_3 = 0.0402$. The optimum solution in terms of EH became $\beta^* = 0.913$ and $R^*_{load} = 18.333\ \Omega$, $I_2 = 0.0689$, and $I_3 = 0.0551$. The maximum $I_2$ and $I_3$ values were much smaller than those in Table 4, indicating that a significant portion of the energy was already dissipated by the primary damping.

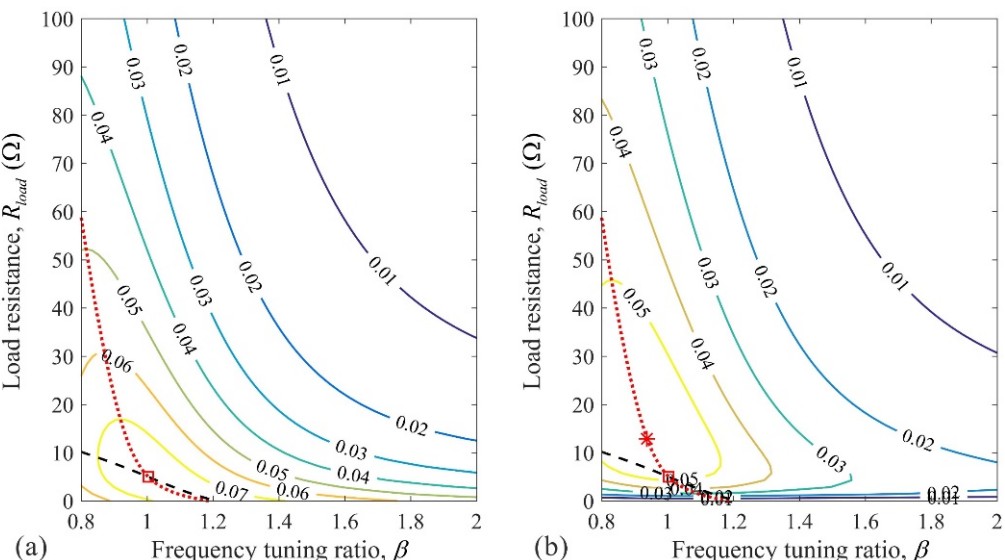

**Figure 9.** Contour plots of $I_2$ (**a**) and $I_3$ (**b**) superimposed with Equation (30)'s solution (black dashed line) and Equation (31)'s solution (red dotted line) when $\mu = 0.08$ and $\zeta_p = 0.15$.

## 4. Validation by Simulation and Experiment with the Band-Limited White Noise

It is not possible to validate the optimum results obtained in Sections 2 and 3 as they are based on an ideal white noise excitation. Using the band-limited white noise base motion, both the numerical simulation and experiment were conducted to validate the general trends revealed in Sections 2 and 3. For this purpose, an apparatus was developed. Figure 10a shows the schematic of the developed apparatus, while in Figure 10b, a photo of the experimental set-up is shown. The primary system consisted of a 3-D printed block used as the primary mass and four steel plates used as the primary spring. The primary system was clamped to a base plate that was fastened to a slip table. The vibration absorber consisted of an aluminum cantilever beam whose free end was attached by a pair of magnets. A portion of each of the magnets moved inside a coil that was held to the fixed stand. Each pair of the magnets and coil formed an electromagnetic energy harvester. As the coils were stationary with respect to the base plate, such configuration resulted in a model B TMD. The magnets and coils were identical to those employed by Yuan et al. [31]. The output leads of the two coils connected in series were fed to the terminals of a resistance substitution box (ELENCO RS-500) that acted as a variable resistive load for the energy harvester. A portion of the cantilever beam was sandwiched between two pieces of 3-D printed plates and was then inserted into the primary block to create a clamed end. By adjusting the length of the 3-D printed plates below the primary block, the length of the cantilever beam was changed so that the frequency tuning ratio could be varied. The relationship between the natural frequency of the TMD and the length of the cantilever beam was obtained by curve-fitting the experimental results,

$$f_a = 18.5118 - 5.2048 \times 10^{-2}L - 1.1139 \times 10^{-3}L^2 + 5.8917 \times 10^{-6}L^3 \qquad (41)$$

where $L$ is the free length of the cantilever beam in millimeters. The length $L$ could be varied from 80 mm to 115 mm that corresponded to $10.327 \geq f_a \geq 6.662$ Hz. Two different mass ratios of $\mu = 0.154$ and $\mu = 0.050$ were considered. Reducing the mass ratio was achieved by adding additional weights to the primary mass block. Table 5 lists the values of the key parameters for the experimental system. Table 6 lists the cantilever beam lengths and their corresponding $\beta$ values. The $\beta$ values used in the experiments are marked in blue.

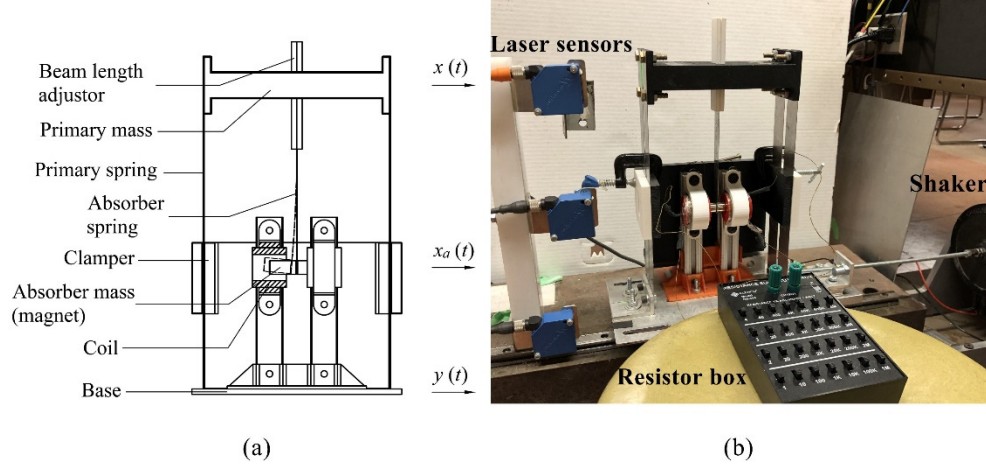

(a) (b)

**Figure 10.** (**a**) schematic of the apparatus; (**b**) photo of the experimental set-up.

**Table 5.** The parameter values of the experimental system.

| Name | Symbol | Value | |
|---|---|---|---|
| Mass ratio | $\mu$ | 0.050 | 0.154 |
| Primary mass | $m$ | 0.957 kg | 0.312 kg |
| Primary stiffness | $k$ | 2125.173 N/m | 931.223 N/m |
| Primary natural frequency | $f_p$ | 7.500 Hz | 8.695 Hz |
| Primary damping ratio | $\zeta_p$ | 0.0056 | 0.0086 |
| Absorber mass | $m_a$ | 0.048 kg | |
| Transduction factor | $\Theta$ | 2.596 Tm | |
| Resistance of coil | $R_{coil}$ | 2.3 $\Omega$ | |

**Table 6.** Different beam lengths and their corresponding frequency tuning ratios.

| Beam Length (mm) | $f_a$ (Hz) | $\beta$ ($\mu$ = 0.050) | $\beta$ ($\mu$ = 0.154) |
|---|---|---|---|
| 115 | 6.662 | 0.888 | 0.766 |
| 111.5 | 6.996 | 0.933 | 0.805 |
| 108 | 7.329 | 0.977 | 0.843 |
| 104.5 | 7.662 | 1.022 | 0.881 |
| 101 | 7.995 | 1.066 | 0.920 |
| 97.5 | 8.327 | 1.110 | 0.958 |
| 94 | 8.661 | 1.155 | 0.996 |
| 90.5 | 8.994 | 1.200 | 1.034 |
| 87 | 9.399 | 1.253 | 1.081 |
| 83.5 | 9.557 | 1.274 | 1.099 |
| 80 | 10.327 | 1.377 | 1.188 |

Figure 11 shows a flow chart of the experimental system. The shaker system (B&K, 2809) with a slip table was driven by a power amplifier (B&K, 2718). An accelerometer (B&K, 4383) was mounted on the slip table to measure the acceleration of the base. Three Laser distance sensors (Wenglor, CP24MHT80) were used to measure the displacements of the base, the primary mass, and the absorber mass, respectively. A desktop computer equipped with a data acquisition board (dSPACE, dS1104) was used to output the excitation

signal to the power amplifier and collect signals of all the sensors. A Simulink model was developed and downloaded to dSPACE ControlDesk software version 3.2.1 (dSPACE GmbH, Paderborn, Germany) to control the experiment.

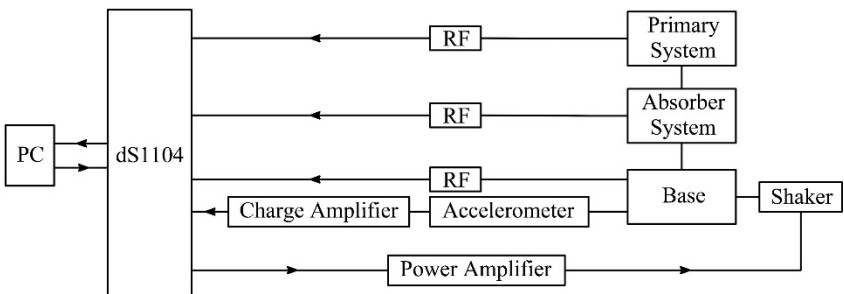

**Figure 11.** Flow chart of the experimental system.

The exciting signal was a band-limited white noise with a frequency range of 4 to 20 Hz. The frequency range was chosen so that the dynamics of the combined system was fully covered. Figure 12 shows the base acceleration measured by the accelerometer and its power spectrum. The PSD of the base acceleration was found to be $S_0 = 1.519 \times 10^{-4}$ (m/s$^2$)$^2$s/rad. For each of the two $\mu$ values, the experiments were conducted in the following manner. The absorber beam length was varied in a step of 3.5 mm. For each of the absorber beam lengths, the experiments were conducted by varying load resistances in the following increments: 0.5 $\Omega$, 1.5 $\Omega$, 3.5 $\Omega$, 7.5 $\Omega$, 10.5 $\Omega$, 20 $\Omega$, 40 $\Omega$, 70 $\Omega$, 100 $\Omega$, 150 $\Omega$, 200 $\Omega$, 400 $\Omega$, and infinity. The infinite load resistance was achieved by an open circuit. The velocities of the primary mass and absorber mass were obtained by differentiating the measured displacements numerically. To alleviate the noise amplification problem in numerical differentiation, the output displacement signals from the sensors were first processed by a low-pass filter with a cut-off frequency of 80 Hz. The filtered signals were then interpolated using a cubic spline approximation. The mean squared relative velocities of the primary mass and absorber mass were computed by

$$E(V^2) \approx \frac{1}{T} \int_0^T \dot{x}_a^2(t)dt \tag{42}$$

and

$$E(V_a^2) \approx \frac{1}{T} \int_0^T \dot{x}_a^2(t)dt \tag{43}$$

, respectively, where $T$ is the duration of the simulation. Then the indices $I_1$, $I_2$, and $I_3$ were computed using Equations (12), (13), and (40), respectively. To improve the smoothness of the results, 10 sets of experiments were conducted for each of the $\beta$ and $\zeta_a$ combinations, and the average values of $I_1$, $I_2$, and $I_3$ were computed. The performance of the absorber in terms of VS and EH was evaluated by examining the values of $I_2$ and $I_3$, respectively.

Figures 13–16 compare the experimental results and the simulation with a mass ratio of $\mu = 0.050$. To make the curves more distinct and easier to identify, only five curves are shown in the 2D plots. All the figures showed that the experimental results matched in general with the simulation ones. The following observations can be made from Figure 13. With a small $\beta$ value, $I_1$ decreased dramatically to a minimum value, then increased slowly as $\zeta_a$ increased. As $\beta$ changed from 0.8 to 1.2, the increasing trend of $I_1$ vs. $\zeta_a$ after reaching a minimum value became less significant, and the overall trend of $I_1$ gradually changed to monotonically decreasing. In the simulation, the minimum value of $I_1$ was achieved when $\beta^* = 1.022$ and $\zeta_a^* = 0.119$, while in the experiment, the minimum value of $I_1$ was achieved when $\beta^* = 1.066$ and $\zeta_a^* = 0.155$. Figure 14 shows that the trend of $I_2$ was opposite to that of $I_1$. When $\beta$ increased from 0.8 to 1.2, with an increase in $\zeta_a$, the trend of $I_2$ first increased to a maximum value and decreased. The maximum $I_2$ value occurred around $\beta^* = 0.977$ and $\zeta_a^* = 0.068$ in the simulation and around $\beta^* = 1.022$ and $\zeta_a^* = 0.155$ in the experiment. Both

the experimental and the simulation results of $I_1$ and $I_2$ confirmed the theory that the sum of $I_1$ and $I_2$ should equal a constant, as proven in Appendix A.

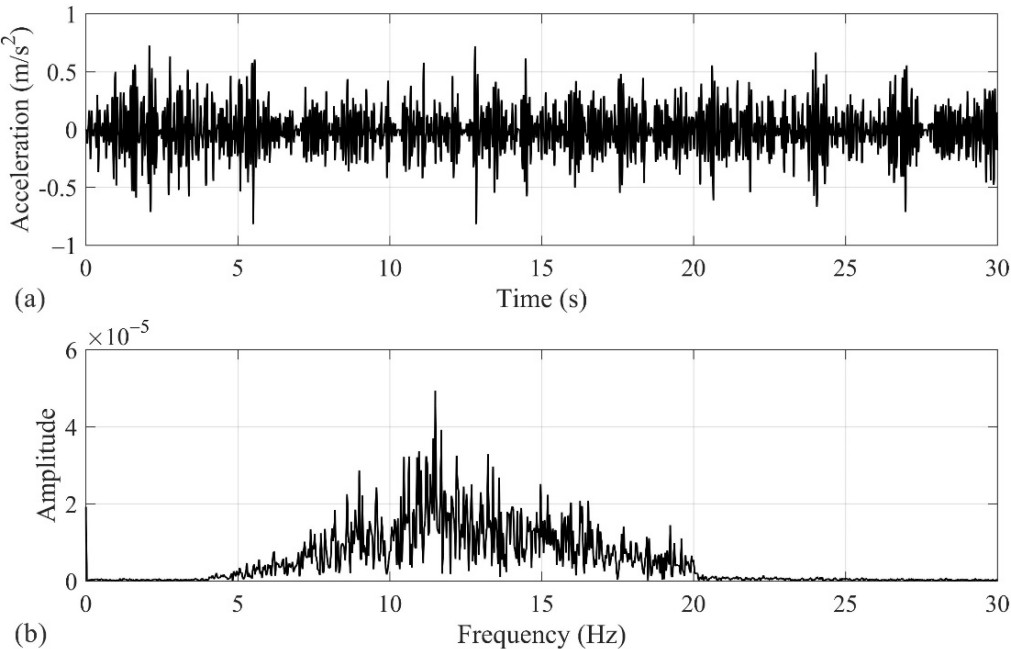

**Figure 12.** Base excitation signal: (**a**) Acceleration; (**b**) FFT spectrum.

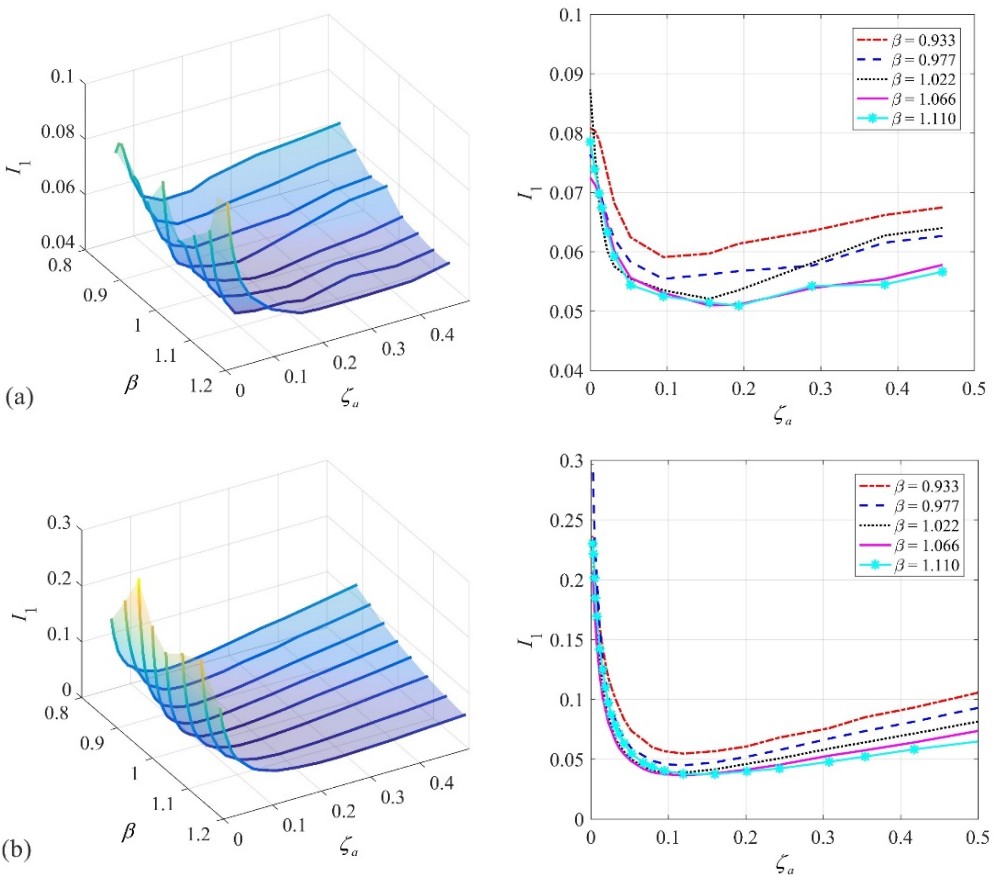

**Figure 13.** 3D and 2D plots of $I_1$ with $\mu = 0.050$: (**a**) experimental result; (**b**) simulation result.

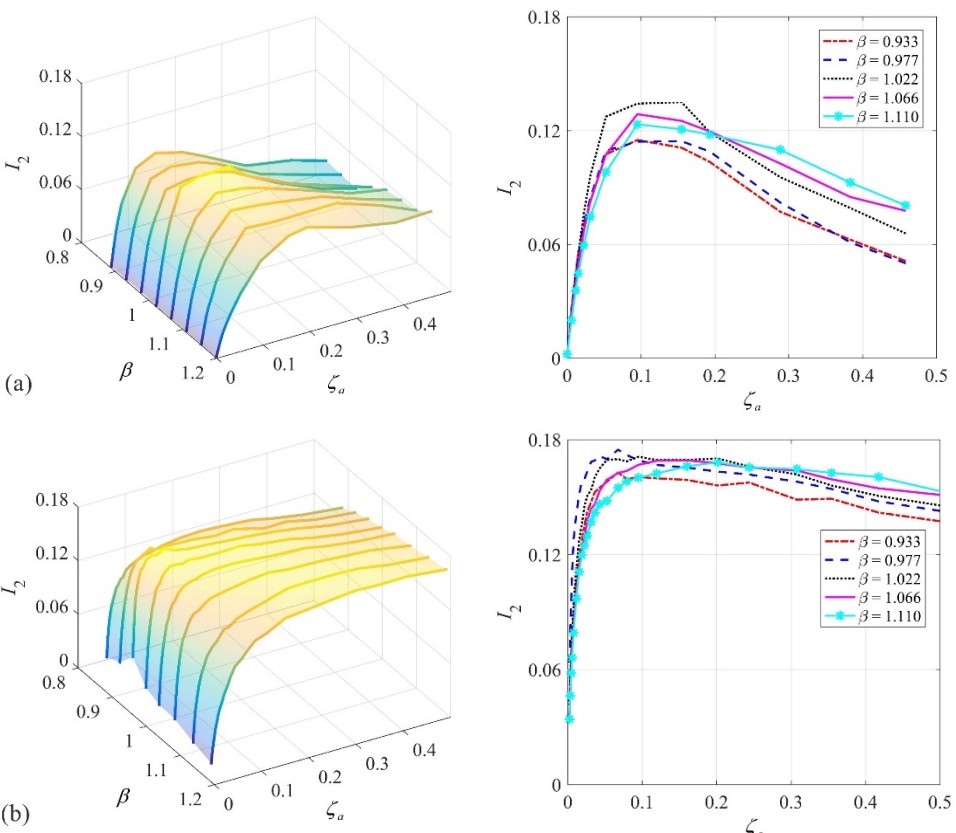

**Figure 14.** 3D and 2D plots of $I_2$ with $\mu = 0.050$: (**a**) experimental result; (**b**) simulation result.

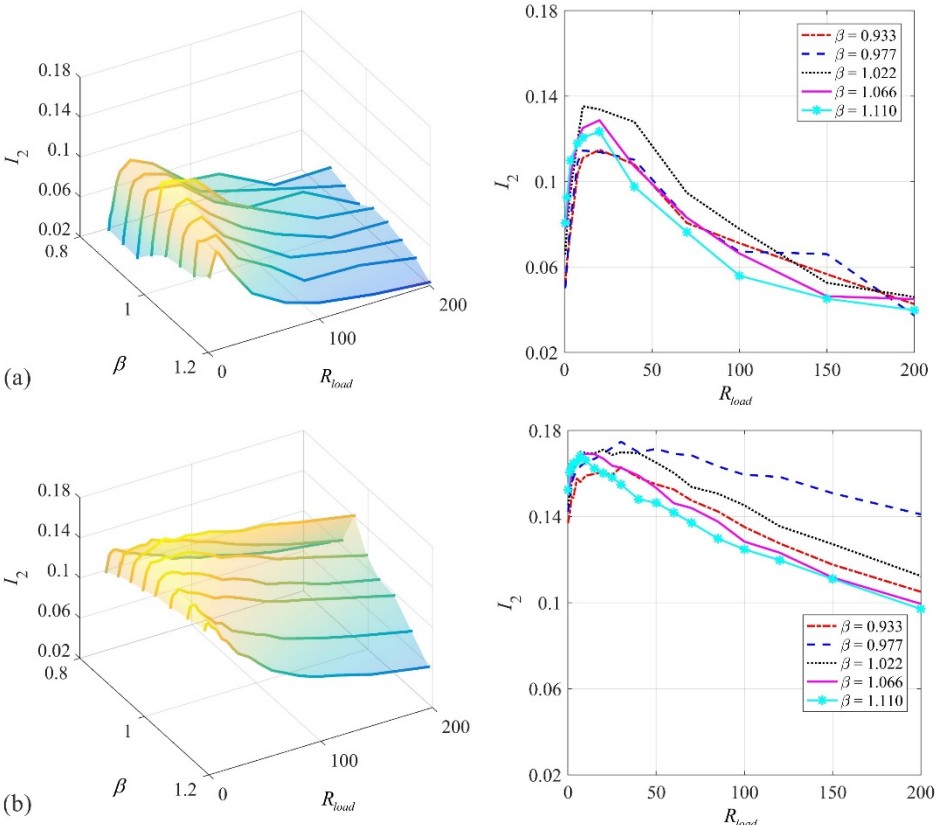

**Figure 15.** 3D and 2D plots of $I_2$ with $\mu = 0.050$: (**a**) experimental result; (**b**) simulation result.

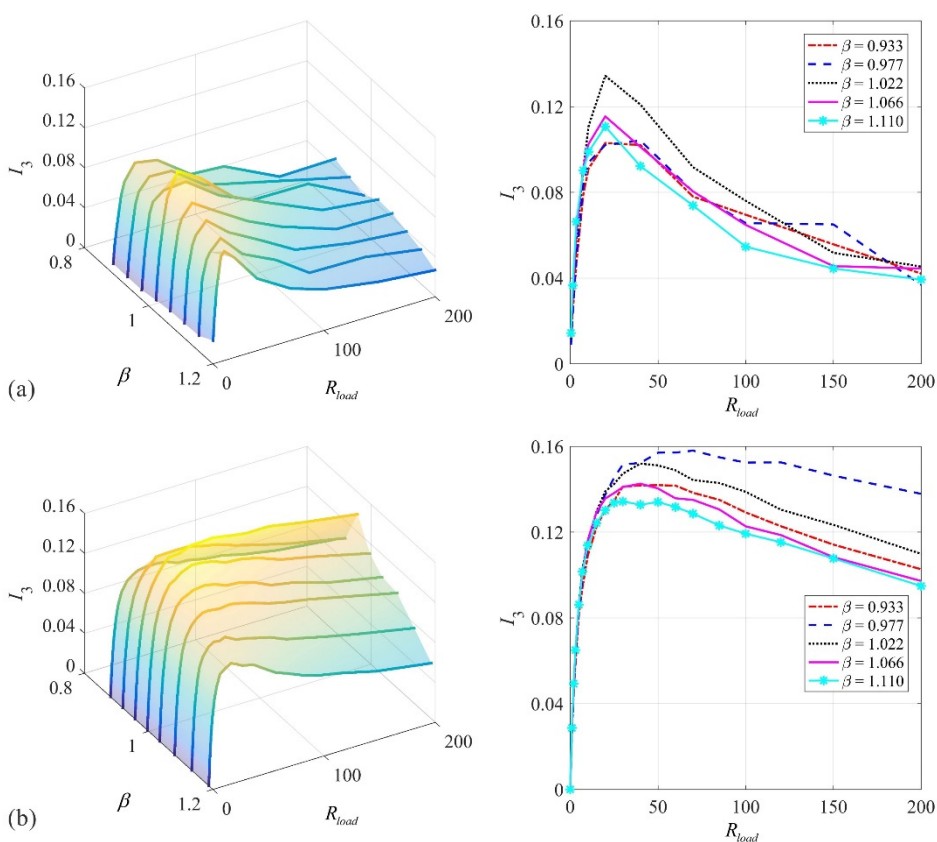

**Figure 16.** 3D and 2D plots of $I_3$ with $\mu = 0.050$: (**a**) experimental result; (**b**) simulation result.

Figures 15 and 16 present the indices $I_2$ and $I_3$ as the function of $\beta$ and $R_{load}$. It can be seen that the maximum $I_2$ value occurred around $\beta^* = 0.977$ and $R^*_{load} = 30.0\ \Omega$ in the simulation and around $\beta^* = 1.022$ and $R^*_{load} = 10.0\ \Omega$ in the experiment, while the maximum $I_3$ value occurred around $\beta^* = 0.977$ and $R^*_{load} = 50.0\ \Omega$ in the simulation and around $\beta^* = 1.022$ and $R^*_{load} = 20.0\ \Omega$ in the experiment. The above observations revealed that in both the simulation and the experiment, reducing the load resistance results in better VS and poorer EH performance. This confirms the aforementioned trade-off between energy dissipated by the absorber damping and the energy harvested by the load resistor.

Figures 17–20 compare the experimental and the simulation results with the mass ratio of $\mu = 0.154$. Comparing Figures 17 and 18 with Figures 13 and 14, the trends of $I_1$ vs. $\zeta_a$ and $I_2$ vs. $\zeta_a$ became more consistent with an increase in $\beta$. There was no local minimum or maximum value for $I_1$ and $I_2$. As shown in Figures 19 and 20, the maximum $I_2$ value occurred around $R^*_{load} = 15.0\ \Omega$ in the simulation and around $R^*_{load} = 10.0\ \Omega$ in the experiment, while the maximum $I_3$ value occurred around $R^*_{load} = 40.0\ \Omega$ in the simulation and around $R^*_{load} = 20.0\ \Omega$ in the experiment. With an increase in the mass ratio $\mu$, the trade-off between the dissipated power and harvested power became more severe for the simulation results than for the experimental ones. The discrepancies between the experimental and the simulation results may mainly be attributed to the two factors. First, the base excitation of the experiment was generated by sending a band-limited white noise signal to the shaker. It is expected that due to the shaker's dynamics, the actual base motion was somewhat different from the input signal used in the simulation. Second, in the experiment, the velocities of the primary mass and the absorber mass were obtained by numerically differentiating the measured displacement signals. Although the measured signals were filtered by a low-pass filter and interpolated with cubic spline approximation to alleviate the noise effect on numerical differentiation, the obtained velocities were not exactly the same as those from the numerical simulation. It is worth noting that the purpose of the experiment was to demonstrate the general trend of the system behaviors revealed

in the analytical result and the numerical simulation. Overall, the analytical predictions and the experimental results are in good agreement.

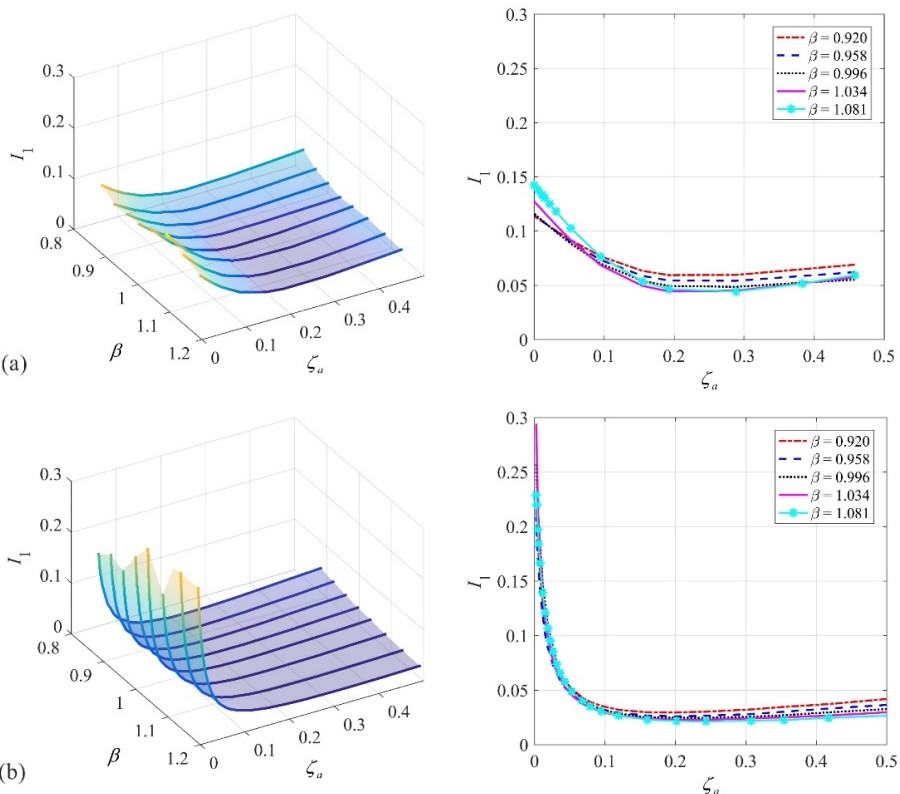

**Figure 17.** 3D and 2D plots of $I_1$ with $\mu = 0.154$: (**a**) experimental result; (**b**) simulation result.

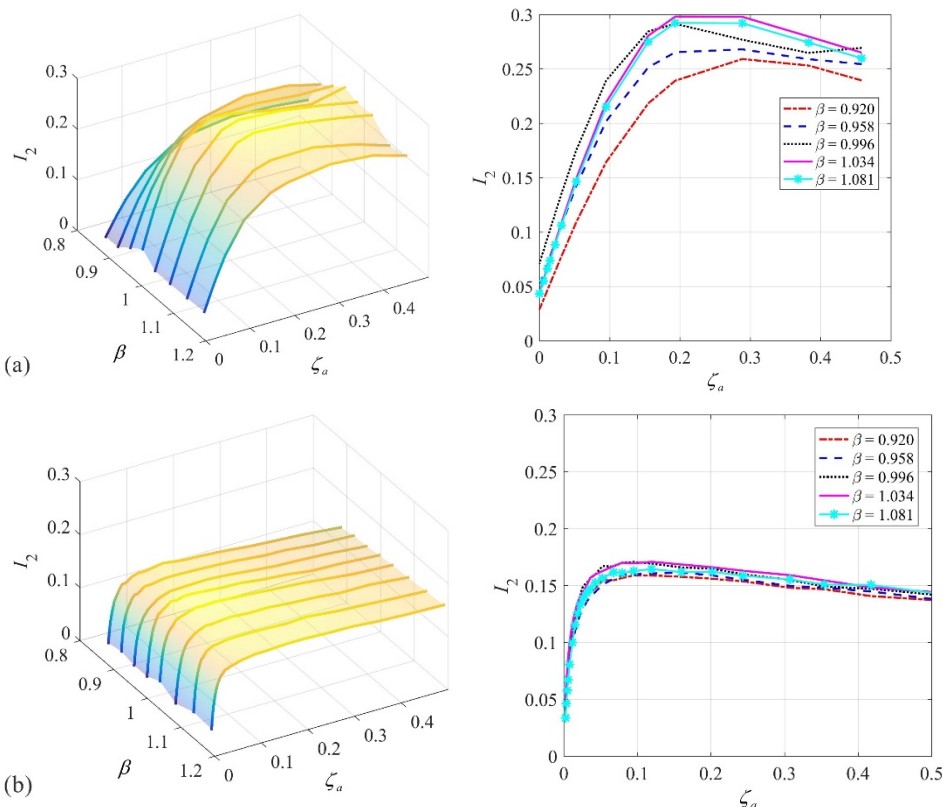

**Figure 18.** 3D and 2D plots of $I_2$ with $\mu = 0.154$: (**a**) experimental result; (**b**) simulation result.

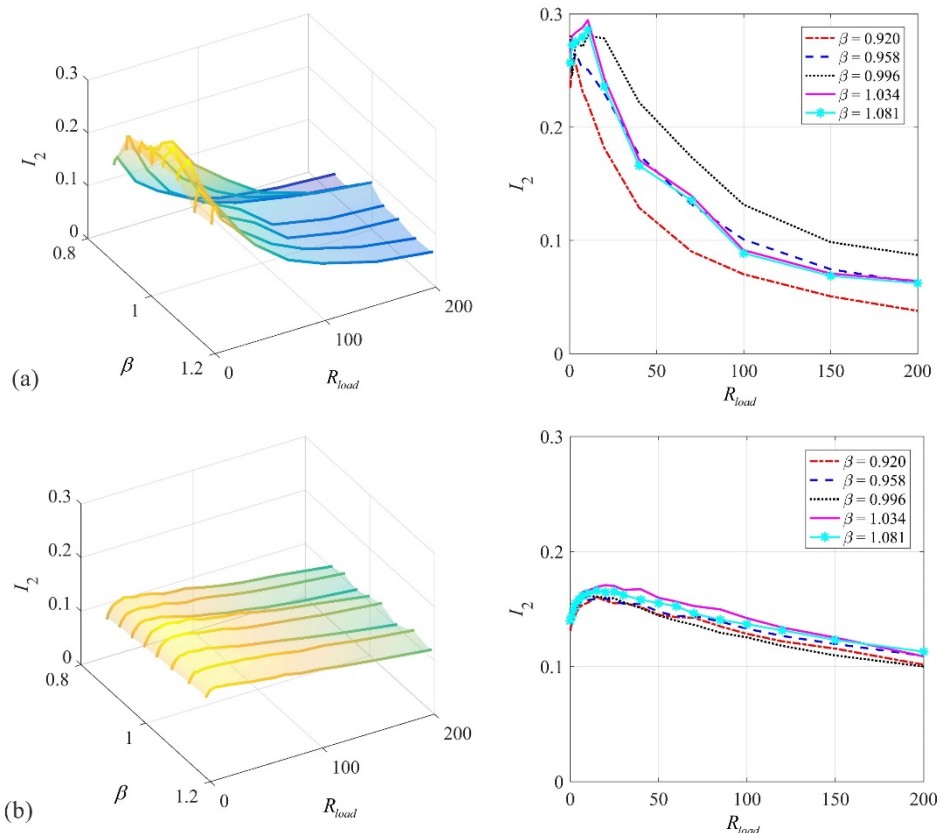

**Figure 19.** 3D and 2D plots of $I_2$ with $\mu = 0.154$: (**a**) experimental result; (**b**) simulation result.

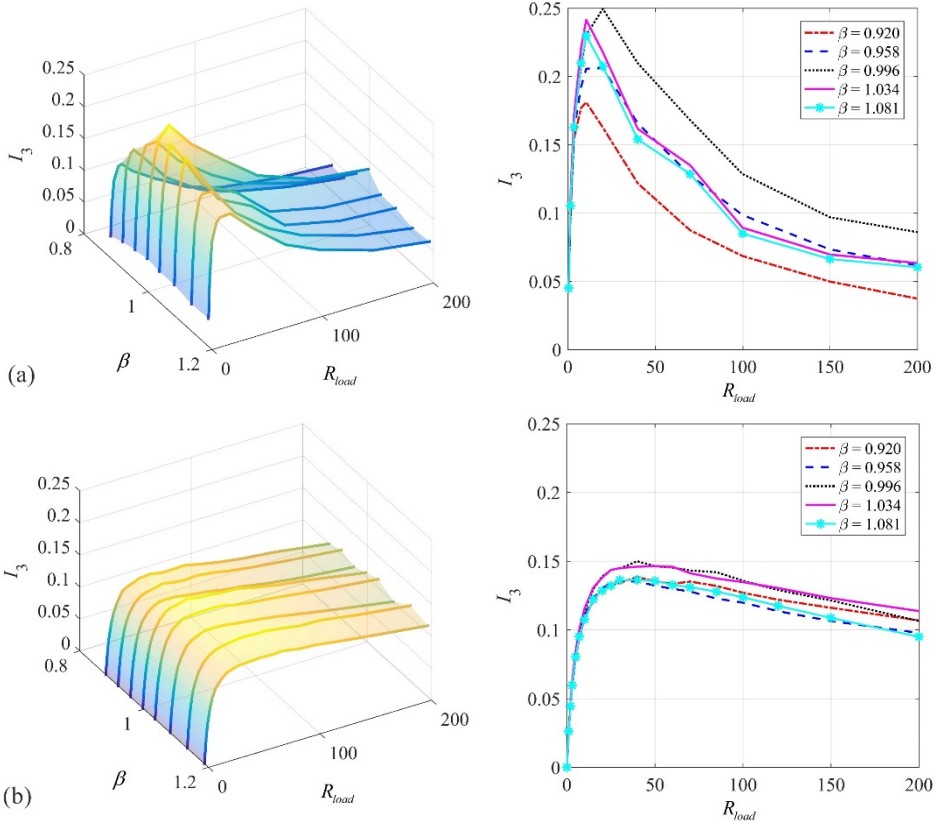

**Figure 20.** 3D and 2D plots of $I_3$ with $\mu = 0.154$: (**a**) experimental result; (**b**) simulation result.

## 5. Conclusions

Optimization of a non-traditional vibration absorber or model B TMD was investigated. An electromagnetic device was used as the absorber damper for a dual purpose of vibration suppression (VS) and energy harvesting (EH) in a primary system subjected to random base excitation. In terms of VS, the objective of optimization was to minimize an index $I_1$ that measured the power dissipated by the primary damper and maximize an index $I_2$ that measured the power dissipated by the absorber damper. It was shown that when the primary system is undamped, the power dissipated by the absorber damper remains a constant. To further investigate the case of the undamped primary system, an alternate performance index $I_p$ was defined to measure the vibration energy of the primary system. It was found that when the mass ratio exceeded 0.142, no local minimum point of $I_p$ existed. For the mass ratio equal to or less than 0.142, with an increase in the mass ratio, the optimum frequency tuning ratio and absorber damping ratio increase as well. For the damped primary system, the minimization of the power dissipated by the primary damping was equivalent to the maximization of the power dissipated by the absorber damper. The optimum frequency tuning ratio and absorber damping ratio were derived when there was a minimum or maximum point. The sensitivity of VS performance was investigated by detuning the optimum tuning frequency ratio or the optimum absorber damping ratio by ±10%. The results showed that the $I_1$ value is more sensitive to detuning if the primary damping is low. For a given primary damping ratio, the $I_1$ value is more sensitive to the variation in the frequency tuning ratio than to the variation in the absorber damping ratio.

In terms of EH, the objective of optimization was to maximize an index $I_3$ that measured the power harvested by the load resistor. The trade-off issue between VS and EH was investigated. It was found that for a given mass ratio and primary damping ratio, the optimum frequency tuning ratio required to maximize VS was slightly higher than that required to maximize the harvested power. However, the optimum load resistance followed a reverse relation. An increase in the mass ratio worsened the trade-off situation. On the other hand, an increase in the primary damping had little effect on the trade-off matter.

An apparatus was developed to validate the analytical results. The developed apparatus allowed both the frequency tuning ratio and the absorber damping ratio to be varied manually. The combined system was excited with a band-limited white noise. The powers dissipated by the primary damping, the power dissipated by the absorber damper, and the power harvested by the load resistor were found experimentally and numerically for the apparatus with two different mass ratios, respectively. The experimental results showed a general agreement with the numerical ones. In particular, the minimization of the power dissipated by the primary damping was equivalent to the maximization of the power dissipated by the absorber damper, and the trade-off between the power dissipated by the absorber damper and the power harvested by the load resistor became worse when the mass ratio increases. These results agree in general with the trends predicted by the analysis.

**Author Contributions:** Conceptualization, M.Y., K.L. and A.S.; methodology, M.Y. and Y.J.; software, M.Y.; validation, Y.J., K.L. and A.S.; formal analysis, M.Y. and Y.J.; investigation, M.Y. and Y.J.; resources, K.L. and A.S.; data curation, M.Y. and Y.J.; writing—original draft preparation, M.Y.; writing—review and editing, Y.J., K.L. and A.S.; visualization, M.Y. and Y.J.; supervision, K.L. and A.S.; project administration, K.L. and A.S.; funding acquisition, K.L. and A.S. All authors have read and agreed to the published version of the manuscript.

**Funding:** This research was financially supported by Natural Sciences and Engineering Council of Canada Discovery Grants (RGPIN/04419-2016, RGPIN/06157-2015).

**Conflicts of Interest:** The authors declare no conflict of interest.

## Abbreviations

| | |
|---|---|
| $m$, $k$, and $c$ | mass, stiffness, and damping coefficient of the primary system |
| $m_a$, $k_a$, and $c_a$ | mass, stiffness, and damping coefficient of the absorber system |
| $y$ | displacement of the base |
| $x$, and $x_a$ | displacement of the primary mass and the absorber mass relative to the base |
| $\omega_p$ and $\omega_a$ | natural frequency of the primary system and the absorber system |
| $\mu$ | mass ratio |
| $\zeta_p$ and $\zeta_a$ | damping ratio of the primary system and the absorber system |
| $Y$ and $\omega$ | amplitude and frequency of the base motion |
| $X$ and $X_a$ | steady-state displacement amplitude of the primary mass and the absorber mass |
| $r$ | frequency ratio |
| $\beta$ | frequency tuning ratio |
| $V$ and $V_a$ | steady-state velocity amplitude of the primary mass and the absorber mass |
| $G$ and $G_a$ | velocity FRF of the primary mass and the absorber mass |
| $S$ | PSD |
| $E$ | expectation value |
| $I_1$, $I_2$, $I_p$, $I_3$, and $I_4$ | performance indices |
| $\Theta$ | transduction factor |
| $R_{coil}$ and $R_{load}$ | resistance of one coil and the load resistor |
| $p_e$ and $p_{load}$ | power dissipated by the absorber damper and harvested by the load resistor |
| $i$ | current |
| $f$ | ratio of harvested power to power dissipated by the absorber damper |
| $E_{load}$ | accumulated energy harvested by the load resistor |
| $S_0$ | PSD of the band-limited white noise |
| $T$ | duration of simulation |
| $L$ | length of the cantilever beam |

## Appendix A

The constants in Equations (16) and (17)

$A_1 = (1 + \mu)(\zeta_a + \mu\zeta_p)$, $A_2 = 4\zeta_p\zeta_a^2(1 + \mu) + 4\zeta_a(\zeta_p^2 + \mu\zeta_a^2) - 2\zeta_a$,
$A_3 = 4\zeta_a^2(\zeta_p + \zeta_a) + \zeta_a$, $A_4 = 4\zeta_p(\mu\zeta_a + \zeta_p)(\zeta_p + \zeta_a) + 2\mu\zeta_p$, $A_5 = 4\zeta_p\zeta_a(\zeta_p + \zeta_a) + \zeta_p$,
$B_1 = (\zeta_p + \mu\zeta_a)(\zeta_a + \mu\zeta_p)$, $B_2 = 4\zeta_a^2\zeta_p^2(1 + \mu) + 4\zeta_p\zeta_a(\zeta_p^2 + \mu\zeta_a^2) + 2\zeta_p\zeta_a(\mu - 1)$,
$B_3 = \zeta_p\zeta_a + 4\zeta_p\zeta_a^2(\zeta_p + \zeta_a)$

The proof that the sum of $I_1$ and $I_2$ is a constant is given below.

$$I_1 + I_2 = \frac{\zeta_p}{4}\frac{A_1\beta^4 + A_2\beta^2 + A_3}{B_1\beta^4 + B_2\beta^2 + B_3} + \frac{\zeta_a\mu}{4}\frac{A_1\beta^4 + A_4\beta^2 + A_5}{B_1\beta^4 + B_2\beta^2 + B_3}$$
$$= \frac{1}{4}\frac{(\zeta_p A_1 + \zeta_a\mu A_1)\beta^4 + \left(\zeta_p A_2 + \zeta_a\mu A_4\right)\beta^2 + \zeta_p A_3 + \zeta_a\mu A_5}{B_1\beta^4 + B_2\beta^2 + B_3}$$

Using the terms defined above, the following can be derived.

$$\zeta_p A_1 + \zeta_a\mu A_1 = (\mu + 1)(\zeta_p + \mu\zeta_a)(\zeta_a + \mu\zeta_p) = (\mu + 1)B_1$$

$$\zeta_p A_2 + \zeta_a\mu A_4 = \zeta_p[4\zeta_p\zeta_a^2(\mu + 1) + 4\zeta_a(\zeta_p^2 + \mu\zeta_a^2) - 2\zeta_a] + \zeta_a\mu[4\zeta_p(\mu\zeta_a + \zeta_p)(\zeta_p + \zeta_a) + 2\mu\zeta_p]$$
$$= 4\zeta_p^2\zeta_a^2(\mu + 1) + 4\zeta_p\zeta_a(\zeta_p^2 + \mu\zeta_a^2) - 2\zeta_p\zeta_a + 4\mu\zeta_p\zeta_a(\mu\zeta_a + \zeta_p)(\zeta_p + \zeta_a) + 2\mu^2\zeta_p\zeta_a$$

$$= (\mu + 1)[4\zeta_p^2\zeta_a^2(\mu + 1) + 4\zeta_p\zeta_a(\zeta_p^2 + \mu\zeta_a^2) + 2\zeta_p\zeta_a(\mu - 1)] = (\mu + 1)B_2$$
$$\zeta_p A_3 + \zeta_a\mu A_5 = \zeta_p[4\zeta_a^2(\zeta_p + \zeta_a) + \zeta_a] + \zeta_a\mu[4\zeta_p\zeta_a(\zeta_p + \zeta_a) + \zeta_p]$$
$$= 4\zeta_p\zeta_a^2(\zeta_p + \zeta_a) + \zeta_p\zeta_a + 4\mu\zeta_p\zeta_a^2(\zeta_p + \zeta_a) + \mu\zeta_p\zeta_a$$
$$= (\mu + 1)[4\zeta_p\zeta_a^2(\zeta_p + \zeta_a) + \zeta_p\zeta_a] = (\mu + 1)B_3$$

Thus,

$$I_1 + I_2 = \frac{\mu + 1}{4}$$

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
