# Peer review of "Optimization of a Non-Traditional Vibration Absorber for Vibration Suppression and Energy Harvesting"

_vibration, doi:10.3390/vibration5030022_

Round 1

Reviewer 1 Report

1. The main difference of this work from the authors' previous works [Yuan and Liu, YUan, Liu and Sadhu] should be clearly addressed. Several figures are same.

2. What causes the difference between simulation and experimental results shown in Figures 14 -20? When does the error is the largest? Why?

3. As mentioned, the purpose of this work is to supress the vibration as well as the energy harvesting. Therefore, several 3D plots repsenting vibration suppression (index), energy harvesting (index) and several main parameters. Then, potential reasder can understand the eefectiveness of the proposed approach without the trade-off between large vibration reduction and high energy harvesting.

Reviewer 2 Report

Thank you for the intereting article. Below I give some of my criticisms. But not to diminish the value of your work. Perhaps many of my questions are caused by my inquisitiveness. However, I hope that some of my comments will improve the perception of your article by the reader.

 1.      The abstract is too long and looks like an introduction. In my opinion, the abstract of a scientific article should include at least 6-8 sentences. The first two sentences should reveal the relevance of the subject of research. The second two-four sentences should talk about the new approaches used by the authors of the work. And the last two sentences should report the main results of the work.

2. In the introduction part, the last sentence should be explained the novelty of this work.  It is not clear whether there is a novelty in this.

3. The papers on simultaneous energy harvesting and vibration mitigation have been added in the introduction part. The methods for finding the compromise region between both effects should be added (like special dynamics indicators in the pendulum vibration absorber/harvester).

4. The authors assumed linear springs. However, the stiffness is more or less nonlinear. How the nonlinearity of the spring influences this optimization method?

5. The author strong simplified the electrical approach and assumed small coil inductance leads that the transducer factor is treated as electrical damping and east estimated. Please explain how these results change if we include the coil inductance?

6. The authors say, on page: 12,, Equation (38) shows that increase of the load resistance results in an increase of the harvested power’’. However, in many papers, the optimal load resistance is close to the coil resistance (Rc=R_load). how can this be explained?

7. Please explain the local minimum point and how this point has been calculated. This is not clear.

8. Please add the legend in contour plots (left figures) in Figs. 13-20.

9. What is the effectiveness of energy harvesting of models A and B?. How the effectiveness is related to the vibration mitigation effectiveness?

10. The conclusion chapter is too long. In my opinion, only the most important things have been included.
